# UNDERSTANDING PURE CLIP GUIDANCE FOR VOXEL GRID NeRF MODELS

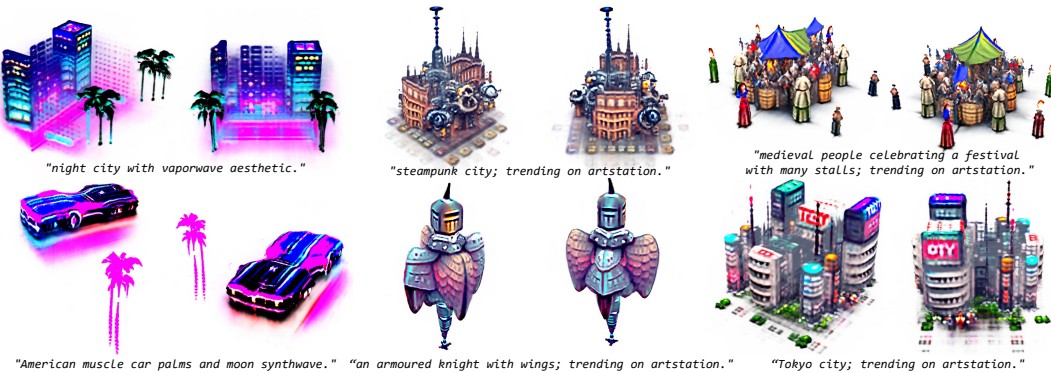

Figure 1: Examples of multi-view images generated from the input prompts with our implicit Vox$_{Imp}$ model trained at resolution $224^2$ and various CLIP models. Our model produces highly detailed 3D representations roughly matching the input text.

## ABSTRACT

We explore the task of text to 3D object generation using CLIP. Specifically, we use CLIP for guidance without access to any datasets, a setting we refer to as pure CLIP guidance. While prior work has adopted this setting, there is no systematic study of mechanics for preventing adversarial generations within CLIP. We illustrate how different image-based augmentations prevent the adversarial generation problem, and how the generated results are impacted. We test different CLIP model architectures and show that ensembling different models for guidance can prevent adversarial generations within bigger models and generate sharper results. Furthermore, we implement an implicit voxel grid model to show how neural networks provide an additional layer of regularization, resulting in better geometrical structure and coherency of generated objects. Compared to prior work, we achieve more coherent results with higher memory efficiency and faster training speeds.

## 1 INTRODUCTION

Text to image generation has seen major recent advances with the release of DALLE (Ramesh et al., 2021) and diffusion models such as DALLE2 (Ramesh et al., 2022), CogView2 (Ding et al., 2022) and Latent Diffusion (Rombach et al., 2022). A natural next step is the task of generating 3D objects from text input. However, supervised methods relying on paired image-text data are less suited to text to 3D generation as large-scale paired text and 3D datasets are less available. Thus, the regime of little to no 3D data training supervision is beneficial. While this might seem daunting, recent work in text to 3D generation showed promising results without using large-scale datasets by bridging the gap using guidance from pretrained vision-language models such as CLIP (Radford et al., 2021).

At the same time, advances in differentiable neural rendering and the development of NeRF (Mildenhall et al., 2020) now allow for direct optimization of a 3D representation to match input images. Combining these approaches with CLIP guidance, we can generate 3D representations from text

---

Supplementary Website: `https://isekaicoder.github.io/ICLR3801-Supplemental/`

directly without paired text–3D data, by optimizing the similarity of the text to the rendered images. Work that leverages CLIP for text to 3D generation can be grouped by the amount of 3D data required. The first set of methods train a generative model on a 3D dataset, and then optimize a mapping network from text to the latent space of the generative model using CLIP guidance and differentiable rendering. The second set of methods utilizes no 3D or text supervision and has access to only the pretrained CLIP model. We refer to this latter regime as *pure CLIP guidance*. Given the scarcity of text–3D pair datasets, we focus on this regime.

A prominent example of the pure CLIP guidance regime is Dream Fields (Jain et al., 2022) which uses Mip-NeRF (Barron et al., 2021) and CLIP to guide the 3D optimization process for every new input text prompt. Unfortunately, this approach requires significant computational resources and exhibits poor quality generation with low-density artifacts when using direct voxel grid optimization (see appendix of original paper). We also find that the quality of the results in Dream Fields is largely attributable to the LiT (Zhai et al., 2022) guidance model. When using the vanilla CLIP models as in our work, results are far worse. Optimizing the CLIP similarity is also prone to adversarial examples where generated images with high similarity according to CLIP have little perceived resemblance to the text description for a human (Liu et al., 2021). Recent text to 3D methods use image-based augmentations as regularization to prevent these issues. However, there has been no systematic study of which of these regularizations matters and how much. In addition, there are several possible design choices for the NeRF and CLIP modules, including the use of explicit voxel grids without any neural networks vs implicit neural representations. We systematically compare these and other factors that impact generation quality, and show that it is possible to generate highly detailed 3D representations with voxel grids alone.

Our main contributions are: 1) We conduct a systematic study of augmentations and their effect on text to 3D generation results with pure CLIP guidance; 2) We compare different CLIP backbones for guidance as well as model ensembles for finer 3D object detail; 3) We compare the regularization effects on geometry of explicit vs implicit voxel grids; and 4) We demonstrate generation of high-resolution grids using CLIP guidance only.

## 2 RELATED WORK

**Text to Image.** Recent text to image generation work has shown impressive results, from autoregressive methods (Ramesh et al., 2021; Ding et al., 2021; Yu et al., 2022) to diffusion methods (Nichol et al., 2021; Ramesh et al., 2022; Rombach et al., 2022; Saharia et al., 2022; Ding et al., 2022). However, these methods require significant computational resources, and data supervision which is harder to obtain in large quantities in the case of 3D objects and corresponding text prompts. To alleviate this problem, several works use CLIP and a pretrained image generator for image generation or manipulation without explicit supervision of corresponding pairs. VQGAN-CLIP (Crowson et al., 2022) passes a randomly initialized image through a pretrained VQGAN encoder to get the latent vector. The latent vector is then fed through the decoder, and after applying several augmentations, CLIP similarity is calculated to use as a loss to optimize the latent vector. Fuse Dream (Liu et al., 2021) shows that CLIP similarity scores are prone to adversarial attacks and that applying Diff Augment (Zhao et al., 2020) results in more robust CLIP scores that can be used for optimization.

**Text to 3D with CLIP Guidance.** Initial work in text to 3D shape generation relied on paired 3D shape and text data for supervised training of joint 3D-text embedding spaces (Chen et al., 2018; Jahan et al., 2021; Liu et al., 2022). Large pretrained image-text embeddings and differentiable rendering led to recent work demonstrating that CLIP and NeRF enable 3D object generation (Sanghi et al., 2022; Jain et al., 2022), manipulation (Michel et al., 2022; Wang et al., 2022; Youwang et al., 2022), and even 3D human animation generation (Hong et al., 2022) without direct supervision of text and 3D corresponding pairs. Here we distinguish between two different levels of supervision in these works. Although there are no corresponding text and 3D examples in the first category, a dataset of 3D objects is available. CLIP-Forge (Sanghi et al., 2022) first trains an implicit (occupancy) autoencoder model on the ShapeNet dataset. Then in a second stage, a normalizing flow model is trained with multi-view images and CLIP to project from the CLIP latent space onto the latent space of the autoencoder. CLIP-NeRF (Wang et al., 2022) uses a similar approach by training a conditional NeRF model and a mapping network to predict updates for the conditional codes according to the input text. Text2Mesh (Michel et al., 2022) takes an input base mesh matching

the semantic class of the input text and trains color and displacement prediction for the base mesh according to the text prompt using CLIP similarity as the primary loss. CLIP-Mesh (Khalid et al., 2022) starts from a sphere mesh initialized with random normal and texture maps, and optimizes mesh vertices and maps through differentiable rendering and CLIP similarity between text and current object. Dream Fields (Jain et al., 2022) utilizes a NeRF representation instead to generate views of the 3D object from a set of cameras and uses a similar optimization process. As shown in FuseDream (Liu et al., 2021), CLIP is very prone to adversarial generations. In the first class of methods using 3D dataset supervision, generations are constrained with a prior over 3D objects making the adversarial generation problem less severe. Since no such datasets are present in the second regime, augmentation techniques are utilized to prevent such adversarial generations in the case of Dream Fields and CLIP-Mesh. CLIP-Forge, CLIP-Mesh, and Dream Fields are text-to-3D generation methods, while CLIP-NeRF and Text2Mesh are text-to-3D manipulation methods. Also, Text2Mesh, Dream Fields, and CLIP-Mesh require per-prompt training, while the other works do not require training/tuning for new text prompts during inference. Compared to previous work without 3D supervision, we systematically study different image augmentations, CLIP architectures, and ensembling of CLIP models, and the corresponding impact on generation results.

**Hybrid Representation NeRFs.** In the original NeRF (Mildenhall et al., 2020), a coordinate-based MLP continuously represents a 3D scene compared to traditional explicit representations such as voxel grids. However, slow and memory-intensive computation is required to render even a single image due to the number of forward passes required for sampled points along rays before plugging into volumetric rendering. Other work (Liu et al., 2020; Yu et al., 2021; Garbin et al., 2021; Hedman et al., 2021; Reiser et al., 2021) have utilized a hybrid implicit-explicit representation to speed up rendering. However, such methods still use coordinates and Fourier embeddings to represent the input. More recent work propose learning the input features stored directly in voxel grid structures. Due to learning better input representations, the MLPs used in these models can be smaller, and density values can be used to prune calculations for voxels on the grid. DVGO (Sun et al., 2022) represents the scene as an explicit density grid and learns feature grid along with a view-dependent MLP to model color. TensoRF (Chen et al., 2022) factorizes the voxel grids as vectors and matrices, allowing for a smaller memory footprint. InstantNGP (Müller et al., 2022) utilizes multiresolution hierarchies of hash tables to represent the learned feature grids and another MLP to predict density and color. Plenoxels (Fridovich-Keil et al., 2022) learns a sparse voxel grid of density values and spherical harmonic coefficients for color. ReLU Fields (Karnewar et al., 2022) also utilizes explicit density grids with spherical harmonics for color and shows that using ReLU activation after interpolating query point in the grid result in competitive results. Our work builds on DVGO. In the explicit voxel grid model, the density and color grids are optimized directly with no neural networks. This is akin to learning alpha and RGB values in an image directly in the 2D case. For the implicit version, we use neural networks to predict the density and color grids.

## 3 BACKGROUND

**CLIP.** Radford et al. (2021) implement CLIP as an image encoder $\text{Enc}_I$ and text encoder $\text{Enc}_T$ trained using contrastive losses. The two encoders project image $x_I$ and text $x_T$ into vectors in an aligned latent space $\boldsymbol{v}_I = \text{Enc}_I(x_I) \in \mathbb{R}^{n_z \times 1}$, $\boldsymbol{v}_T = \text{Enc}_T(x_T) \in \mathbb{R}^{n_z \times 1}$. Since the latent space is aligned between two modalities, the similarity between the image and text is given by the cosine similarity $\boldsymbol{v}_I \cdot \boldsymbol{v}_T / \|\boldsymbol{v}_I\|\|\boldsymbol{v}_T\|$. The model is pretrained on a large-scale dataset collected by OpenAI consisting of more than 400 million image text pairs. We refer to different pretrained CLIP models by the architecture of the image encoder such as "ViT-B/32", "ViT-B/16" etc.

**Dream Fields.** Jain et al. (2022) introduced the use of pure CLIP guidance for 3D object generation. For the backbone 3D representation, they use an implicit NeRF representation, specifically Mip-NeRF(Barron et al., 2021) which uses conical frustums over sampled points to calculate an integrated positional encoding (IPE) as opposed to the positional encoding (PE) in the original NeRF. CLIP similarity between the rendered image and input text prompt is used to facilitate training. They use a background augmentation scheme to prevent adversarial generations.

**Direct Voxel Grid Optimization.** In DVGO (Sun et al., 2022), voxel grids are used to represent the density and color in NeRF. The density grid $\boldsymbol{V}^{(\text{density})} \in \mathbb{R}^{1 \times N_x \times N_y \times N_z}$ is modeled explicitly. The color grid is either modeled explicitly as $\boldsymbol{V}^{(\text{rgb})} \in \mathbb{R}^{3 \times N_x \times N_y \times N_z}$, or with a learnable feature grid

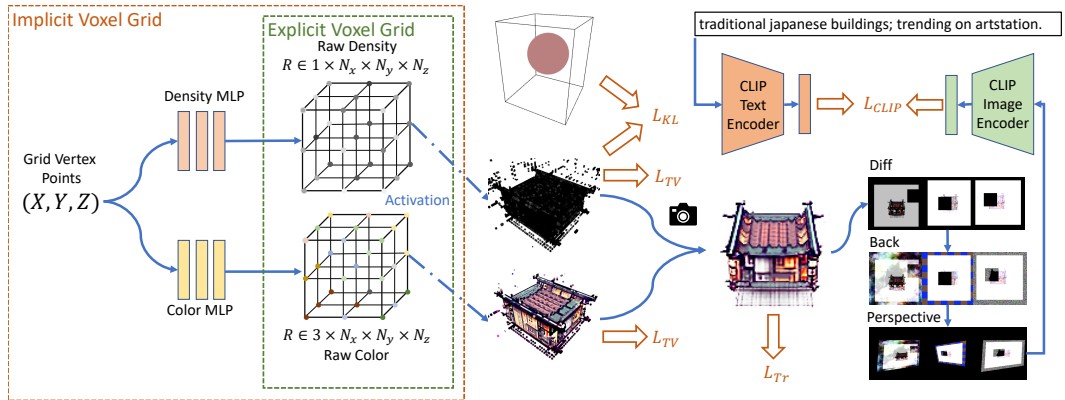

Figure 2: We investigate the effect of different augmentations on text to 3D generation using pure CLIP guidance. We study two radiance field representations: explicit voxel grid Vox$_{\text{Exp}}$ with density and color, and an implicit Vox$_{\text{Imp}}$ model that uses MLPs to predict the density and color. We train our model using CLIP guidance by optimizing the similarity of the CLIP encoded latent vectors of the rendered image and the input text prompt. The rendered image is augmented before CLIP encoding (with DiffAug, BackAug and PerspAug). We introduce KL divergence loss against a spherical prior (red sphere) to enforce a compact scene - the box enclosing the red sphere shows the scene bounds.

$\boldsymbol{V}^{(\text{feat})} \in \mathbb{R}^{D \times N_x \times N_y \times N_z}$ and shallow MLP to predict color for query points implicitly where $D$ is the feature length. The original DVGO had two stages: a coarse stage to locate the object in the grid, and a fine detail stage. There is no ground truth surface in our task, so we do not require a coarse stage. Volumetric rendering follows the quadrature rule (Max, 1995). Concretely, Eq. 1 is used to render the color $\hat{C}(r)$ for a ray $r$ where $\delta_i$ is the interval between adjacent samples, $\alpha_i$ is the alpha compositing value, $K$ is the number of queried points along the ray, and $\text{Tr}_i$ is the transmittance.

$$\hat{C}(r) = \sum_{i=1}^{K} \text{Tr}_i \, \alpha_i c_i, \text{ where } \alpha_i = \text{alpha}(\sigma_i, \delta_i) = 1 - \exp(-\sigma_i \delta_i), \text{ and } \text{Tr}_i = \prod_{j=1}^{i-1}(1 - \alpha_j) \quad (1)$$

The density $\sigma$ and color $c$ are interpolated from voxel grid values at the sample point coordinate and passed through a shallow MLP for implicit feature grids. The sigmoid function is applied to the raw color value. For the density, a post-activation scheme is used with the softplus function on the raw density, $\alpha^{(\text{post})} = \text{alpha}(\text{softplus}(\text{interp}(x, \boldsymbol{V}^{(\text{density})})), \delta)$.

## 4 METHOD

**Model architecture.** For our NeRF model, we implement two variations of the voxel grid representation: Vox$_{\text{Exp}}$, an explicit voxel grid representation, and Vox$_{\text{Imp}}$, an implicit version that uses MLPs to predict the density and color. Vox$_{\text{Exp}}$ explicitly models the two voxel grids consisting of the density $\boldsymbol{V}^{(\text{density})} \in \mathbb{R}^{1 \times N_x \times N_y \times N_z}$ and color $\boldsymbol{V}^{(\text{rgb})} \in \mathbb{R}^{3 \times N_x \times N_y \times N_z}$. Vox$_{\text{Imp}}$ is an implicit coordinate-based MLP voxel grid representation with positional encodings (Mildenhall et al., 2020) of the grid vertex coordinates formulated as $\boldsymbol{V}^{(\text{PE})} \in \mathbb{R}^{L \times N_x \times N_y \times N_z}$, where $L$ is the channel size after positional encoding of grid vertex coordinates. Separate density and color MLPs are applied on the positional encodings to obtain the density and color predictions. We base our voxel grid model implementations on DVGO. However, note that our positional encoding feature grid $\boldsymbol{V}^{(\text{PE})}$ is fixed and not learnable like the DVGO feature grid $\boldsymbol{V}^{(\text{feat})}$ for color. Our overall model illustration can be found in Fig. 2. The trainable parameters for the explicit model are the parameters of the explicit grids and the bias term in the softplus activation of the density value. For the implicit voxel grid the trainable parameters are the density and color MLPs and the bias term in the softplus function. During training we also add progressive scaling of the voxel grid resolution as in DVGO.

**Augmentations.** We study the impact of combining three augmentation schemes: Background augmentation (BackAug) from Dream Fields Jain et al. (2022), Diff augment (DiffAug) Zhao et al. (2020), and perspective augmentations (PerspAug) from Text2Mesh Michel et al. (2022). BackAug consists of alpha compositing checkerboard, textures or gaussian noise backgrounds to the image. DiffAug contains several image augmentations, including color jittering, image translation, and

cutout. Liu et al. (2021) used it for text to image generation and showed it prevents adversarial generations. PerspAug denotes random perspective transformations applied to the image.

**Losses.** We combine the CLIP and transmittance losses introduced in Dream Fields, with losses from DVGO to reduce noise and promote smoothness. In addition, we introduce a spherical prior loss term to encourage a coherent object. For a model parameterized by $\theta$, the CLIP loss (Eq. 2) enforces the cosine similarity between the NeRF generated image $I(\theta, \boldsymbol{p})$ for a camera pose $\boldsymbol{p}$ and the input caption $x_T$ to be high in CLIP space. The transmittance loss (Eq. 3) prevents the scene from being overcrowded by applying a loss when the average transmittance is over the threshold $\tau$ and $\mathrm{Tr}(\theta, \boldsymbol{p})$ is the transmittance image.

$$\mathcal{L}_{\mathrm{CLIP}}(\theta, \boldsymbol{p}, x_T) = -\mathrm{Enc}_I(I(\theta, \boldsymbol{p}))^\top \mathrm{Enc}_T(x_T) \tag{2}$$

$$\mathcal{L}_{\mathrm{Tr}} = -\min(\tau, \mathrm{mean}(\mathrm{Tr}(\theta, \boldsymbol{p}))) \tag{3}$$

To encourage centered objects and uniform size, we introduce a spherical prior (Eq. 4) where the probability is 1 for coordinates $\boldsymbol{q}$ within a sphere of radius 1. We calculate the KL divergence between the spherical prior and the density voxel grid with the sampled point coordinates $\boldsymbol{q}$ from grid vertices (Eq. 5). This loss serves the same purpose as ray shifting in Dream Fields. However, it is not trivial to shift and scale the voxel grid directly. Therefore, we promote centering and uniform size through this loss instead.

$$P_{\mathrm{sphere}}(\boldsymbol{q}) = \begin{cases} 1, & \text{if } \|\boldsymbol{q}\|_2^2 \leq 1 \\ 0, & \text{otherwise} \end{cases} \tag{4}$$

$$\mathcal{L}_{\mathrm{KL}_s} = \sum_{\boldsymbol{q}} P_{\mathrm{sphere}}(\boldsymbol{q}) \log \left( \frac{P_{\mathrm{sphere}}(\boldsymbol{q})}{\alpha^{(\mathrm{post})}(\boldsymbol{q}, \boldsymbol{V}^{(\mathrm{density})})} \right) \tag{5}$$

In our model, we enable the ensembling of different CLIP models by adding a second similarity loss using a different CLIP model as shown in Eq. 6, where $\mathrm{Enc}_{I_2}$ and $\mathrm{Enc}_{T_2}$ are the image and text encoders for the second CLIP model respectively.

$$\mathcal{L}_{\mathrm{CLIP}_2}(\theta, \boldsymbol{p}, x_T) = -\mathrm{Enc}_{I_2}(I(\theta, \boldsymbol{p}))^\top \mathrm{Enc}_{T_2}(x_T) \tag{6}$$

Following DVGO, we also add a total variation loss $\mathcal{L}_{\mathrm{TV}}$ to reduce noise and promote smoothness and a background entropy loss $\mathcal{L}_\sigma$ to encourage density values be either 0 or 1. Our complete loss is shown in Eq. 7, where the $\lambda$s are the weights for the different loss terms.

$$\mathcal{L}_{\mathrm{Total}} = \mathcal{L}_{\mathrm{CLIP}} + \lambda_{\mathrm{Tr}}\mathcal{L}_{\mathrm{Tr}} + \lambda_{\mathrm{TV}}\mathcal{L}_{\mathrm{TV}} + \lambda_{\mathrm{KL}_s}\mathcal{L}_{\mathrm{KL}_s} + \lambda_\sigma\mathcal{L}_\sigma + \lambda_{\mathrm{CLIP}_2}\mathcal{L}_{\mathrm{CLIP}_2} \tag{7}$$

We find that during optimization, it is important to schedule the $\mathcal{L}_{\mathrm{TV}}$ and $\mathcal{L}_{\mathrm{KL}_s}$ loss terms so that they are turned off toward the end of the optimization. Please see Appendix A.1 for more discussion and details of other hyperparameter values.

## 5 EXPERIMENTS

### 5.1 IMPLEMENTATION

We sample camera poses following Dream Fields with a fixed elevation of $30°$, azimuth of $360°$ around the scene, and a radius of 4 between the camera and origin. For the augmentations, unless otherwise specified, we turn on all three of them. For DiffAug, we use the same settings as in the original implementation[1]. For BackAug, we use the same JAX implementation from Dream Fields[2]. For PerspAug, we set the distortion to 0.6 with a probability of being applied to 1.0. The augmentations are applied in the order of (1) DiffAug, (2) BackAug, and (3) PerspAug. For most experiments, eight variations are applied for each image with both DiffAug and BackAug and only one for PerspAug, resulting in 64 images per optimization iteration. For larger models trained with CLIP ViT-L/14 and $\mathrm{Vox}_{\mathrm{Imp}}$, only four variations of DiffAug is applied for a given image resulting in 32 images after augmentation. Training is conducted with image resolution $224^2$ during training using the $\mathrm{Vox}_{\mathrm{Exp}}$ model with the CLIP ViT-B/32 model for 15k iterations unless otherwise specified. The MLPs used in our implicit voxel grid are 3 layers deep with hidden layer dimensions of 128.

---

[1] https://github.com/mit-han-lab/data-efficient-gans/blob/master/DiffAugment_pytorch.py
[2] https://github.com/google-research/google-research/tree/master/dreamfields

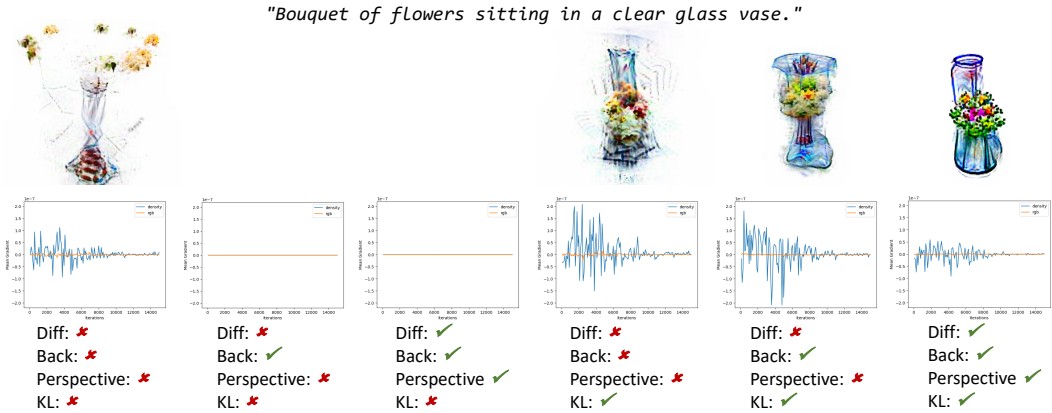

Figure 3: Ablations showing how the prior KL divergence loss with different augmentations affect the generation results of Vox$_{\text{Exp}}$. The chart below the images indicates the mean gradients with respect to the losses for the training iterations. The blue and orange lines represent the mean gradients for the density and color grids respectively. It can be seen that the $\mathcal{L}_{\text{KL}_s}$ loss is essential for the appearance of densities when augmentations are turned on.

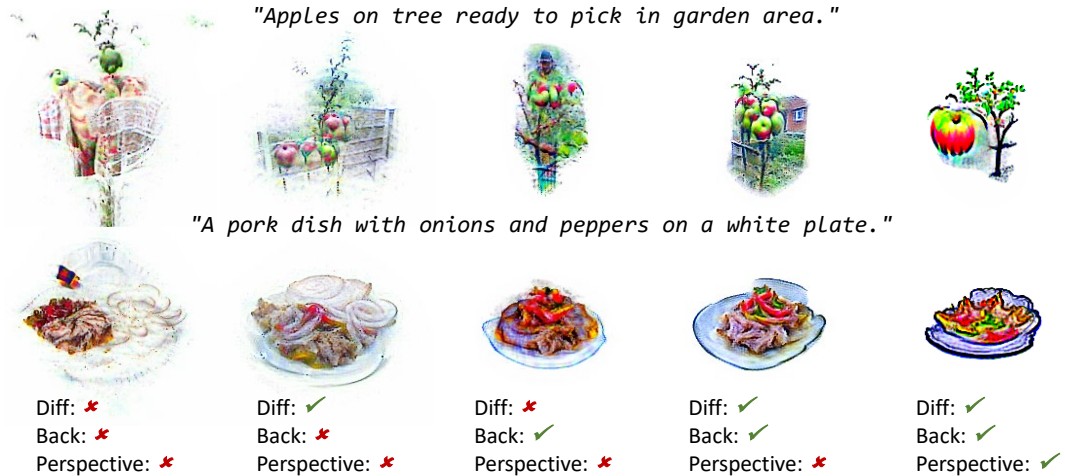

Figure 4: Ablations showing how different augmentations affect the generation results of Vox$_{\text{Exp}}$. We see that different augmentations can have distinct effects on the generation result and that combining them can result in better coherency.

## 5.2 ABLATION

We compare the effect of different augmentations and the KL loss term on Vox$_{\text{Exp}}$. In Appendix A.3, we show the augmentation ablation results on Vox$_{\text{Imp}}$.

**Prior KL Loss Ablation.** We show the effect of the $\mathcal{L}_{\text{KL}_s}$ loss term on the generation results in Fig.3. Blank images indicate that the densities in the entire scene are near zero. When all augmentations and the KL loss is turned off (first column) the generated scene contains floating clouds of non-coherent artifacts. If augmentations are turned on without the KL loss (2nd, 3rd columns), the voxel grid will fail to produce any densities. We hypothesize this is because the CLIP similarity loss landscape is more discontinuous when we add augmentations, making it hard for densities to emerge since there are no gradients in any direction. The gradients are near zero for these cases shown in the plot below the images. When the KL loss is turned on (last 3 columns) the densities are more concentrated and doesn't disappear even when all augmentations are turned on. This could mean that the KL loss term helps to force densities emerge in the beginning allowing for meaningful gradients. Therefore, in addition to regularizing the size and centering of the object generated, the KL loss term also plays a vital role in generating coherent results.

**Augmentation Ablation.** We show the effect of the augmentations: (1) DiffAug, (2) BackAug, and PerspAug and their combinations in Fig. 4. When all three augmentations are turned off (first col-

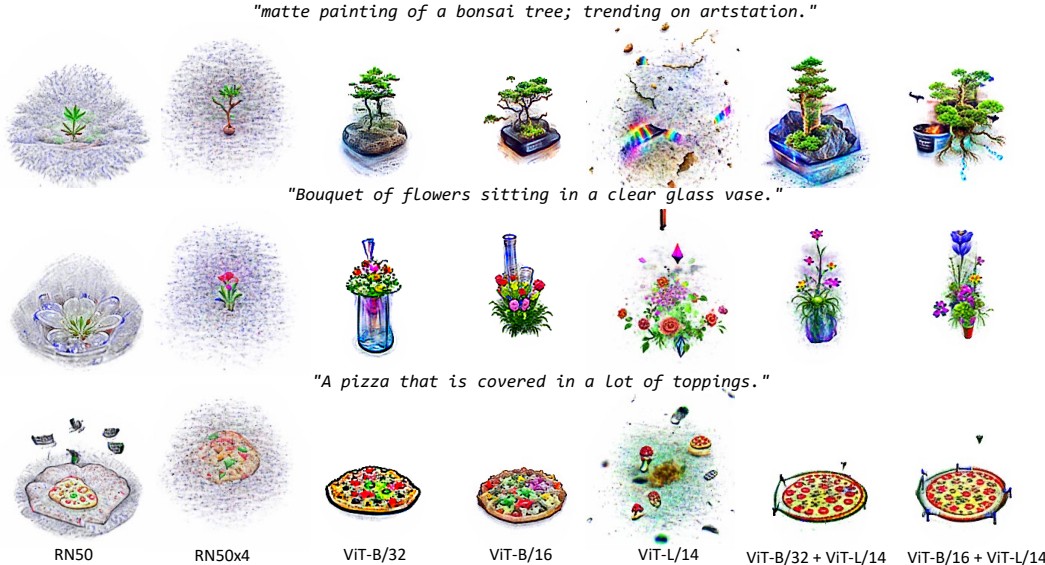

*"matte painting of a bonsai tree; trending on artstation."*

*"Bouquet of flowers sitting in a clear glass vase."*

*"A pizza that is covered in a lot of toppings."*

RN50    RN50x4    ViT-B/32    ViT-B/16    ViT-L/14    ViT-B/32 + ViT-L/14    ViT-B/16 + ViT-L/14

Figure 5: Generation results using different CLIP image encoder architectures. The text below each column indicates the pretrained CLIP architecture used for guidance during training. We see that ensembled CLIP guidance can generate more detailed geometry than using one model.

umn), we can see the results are very noisy and incoherent for the text prompt. With only DiffAug on (col 2), textures start appearing, but the densities are sparsely distributed like point clouds. For results with only BackAug (col 3), the object is condensed and looks more like a contiguous object. When both are turned on (col 4), the objects more clearly match the input prompt. Turning on PerspAug (col 5) helps to get rid of unwanted background textures in the apple tree prompt. However, it may create an overly smooth image or almost cartoonish effect in the plate of food. While we observe that turning off PerspAug has better qualitative results for many prompts, it is infeasible to perform per-prompt tuning, so we leave PerspAug on for following experiments. For the $\text{Vox}_{\text{Imp}}$ model, we observe that the model can produce clean results without background textures even without PerspAug (see Appendix A.3).

### 5.3 DIFFERENT CLIP MODEL EXPERIMENTS

**Backbone Networks.** We compare generation results from 5 pretrained CLIP models with different backbone networks in Fig. 5. Overall the ResNet (He et al., 2016) models (ResNet50, RN50x4) are very noisy, but some features within the generations vaguely resemble the text prompt. It seems that these models have a hard time separating the main subject of the text prompts from other textures. For ViT-B/32 and ViT-B/16 the generations are more coherent and have good separation from unwanted textures compared to the ResNet models. It can also be seen that ViT-B/16 can generate more details, as seen in the *bonsai plant* and *pizza*. This has also been shown in the Dream Fields paper. However, when we use the biggest model, ViT-L/14, the results are very noisy and non-coherent. We hypothesize that as the models increase to a certain point, it becomes easier to find adversarial images within the CLIP space. This could suggest that ViT-L/14 is overfit on the texture features for images, not the objects' overall structure. Discounting the ViT-L/14 model, vision transformers (Dosovitskiy et al., 2021) show much better results than traditional convolutional architectures. We attribute this to the tokenization of image patches, which discretizes the image into regions and may help to disentangle the overall structure and textures of the image.

**CLIP Model Ensemble.** Fig. 5 (last two columns) shows the results of different CLIP ensembles, namely ViT-B/32+ViT-L/14 and ViT-B/16+ViT-L/14. We see that for both ensemble models, we can generate much finer details than using one guidance model alone (see example for the *flowers* and *bonsai plant*). This suggests that using a smaller CLIP model to initialize the generation process helps prevent the bigger model ViT-L/14 from adversarial generations and may even help to generate

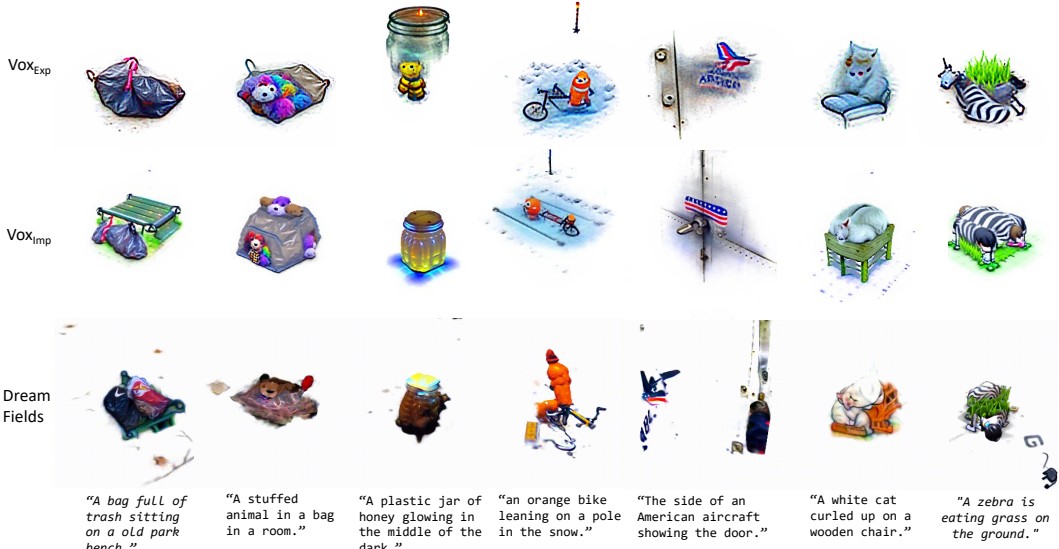

Vox_Exp

Vox_Imp

Dream
Fields

"A bag full of trash sitting on a old park bench." — "A stuffed animal in a bag in a room." — "A plastic jar of honey glowing in the middle of the dark." — "an orange bike leaning on a pole in the snow." — "The side of an American aircraft showing the door." — "A white cat curled up on a wooden chair." — "A zebra is eating grass on the ground."

Figure 6: Our method compared with Dream Fields. All models shown here was trained at resolution $168^2$ for 10k iterations and uses CLIP ViT-B/16 for guidance. We see that the Vox_Imp model generally has the best coherency in terms of both both object texture and geometric structure.

Table 1: We compare the accuracy for R-Precision@1 for our models to that of Dream Fields using different guidance models. R-Precision values in parenthesis (right column) indicate when the CLIP model used to evaluate retrieval is the same as the CLIP model used for optimization.

|  | Retrieval model R-Precision | |
| Method | ViT-B/32 | ViT-B/16 |
| --- | --- | --- |
| Dream Fields (ViT-B/16) | $59.8 \pm 2.8$ | $(93.5 \pm 1.4)$ |
| Vox_Exp (ViT-B/16) | 83.01 | (90.85) |
| Vox_Imp (ViT-B/16) | **88.24** | **(98.69)** |

more fine grain details. This is likely due to the bigger model size of ViT-L/14 as well as the smaller patch sizes it takes in as tokens. However, we also observe that some generation results are worse, such as the *pizza* having fewer details than just using ViT-B/32 or ViT-B/16. We attribute this to the two CLIP models having different maximums for the *pizza*, which may cause conflicting similarity losses and generate less sharp objects.

**Explicit vs Implicit Voxel Grids vs Dream Fields.** We show the results of Vox_Exp, Vox_Imp and Dream Fields in Fig. 6. All three models were trained with image resolution $168^2$ and 10k iterations using the CLIP ViT-B/16 model. Note that we turn off PerspAug for Vox_Imp as it is unnecessary for the implicit model (see Appendix A.3). The Vox_Exp model is able to generate textures that corresponds with the text prompt. However, the geometry does not conform with objects in the real world. The Vox_Imp model in comparison produces objects with better geometrical structure. We see this with the *honey jar*, where the Vox_Imp model generates a cohesive 3D object whereas we see an incomplete jar in the Vox_Exp model. The coherency for Vox_Imp is also better with respect to the text prompt. For example, in the text prompt involving *a bag of trash* and *a park bench*, the Vox_Imp model successfully generates both items whereas there is only a trash bag for Vox_Exp. However, we notice that for complex objects like animals (*cat*, *zebra*) all models fail at producing correct geometry. This is likely due to the limitation of the guidance model used. This failure mode was described as "repeated patterns on multiple sides of the object" by Dream Fields. Overall this is still a very hard task and we find several prompts where the models failed to generate cohesive results as can be seen in the example with the *orange bike* and *aircraft door*. Dream Fields had poor results overall and many generations are unrecognizable with respect to the text prompt. This is in contrast to results shown in their paper using the LiT ViT-B/32 (Zhai et al., 2022) model trained on billions of higher resolutions ($288^2$) captioned images.

For quantitative comparison, we follow Jain et al. (2022) and report the R-Precision@1 retrieval scores for the three models in Tab. 1. We calculate R-Precision@1 retrieval scores for the text and images of the objects on a validation set of 153 text prompts used in the Dream Fields paper. The

images are rendered at held-out poses of elevation $45°$ following Dream Fields. We report the retrieval scores for both ViT-B/16, which the models were optimized on, and a different CLIP model, ViT-B/32. All models are able to achieve high retrieval scores when evaluated on the CLIP model that they were trained on. Overall, we see that the $\text{Vox}_{\text{Imp}}$ model achieves the best quantitative results. The $\text{Vox}_{\text{Exp}}$ model outperforms Dream Fields considerably when evaluated with a different CLIP-model. We note that training at lower resolutions ($168^2$) result in quality degradation compared to models trained at higher resolutions in Fig. 1 and that training at higher resolutions ($224^2$) may result in even better results while still using less memory (see Appendix A.5) compared to Dream Fields.

## 6 DISCUSSION

**3D Representation.** In this paper we mainly explored how pure CLIP guidance affects NeRF models. However, it would be interesting to see if findings here also apply to methods based on explicit representations such as CLIP-Mesh (Khalid et al., 2022). Since they start with a primitive mesh for optimization the possible topology and complexity of possible generations are limited compared to NeRFs. It would be worth exploring whether this helps to regularize CLIP guidance.

**Augmentations.** Augmentations are essential for preventing adversarial generations from CLIP guidance. We showed how different augmentations affect the generation results and that sometimes it can even be harmful to the coherency of the results. This was shown with perspective augmentation. It can help eliminate unwanted textures in the background for some prompts, but can also overly smooth the textures for others. While prior work in this task has not put as much emphasis on testing different augmentations, we think it is important to explore more variations to find the best augmentations for different use cases.

**Guidance Models.** While we may try all sorts of augmentations to improve coherency, the main constricting factor lies within the guidance model such as CLIP. If the guidance model is not able to learn the correct structure and geometry of objects, no augmentations will help bring those out in the results. Therefore, it is important to also improve the guidance models. We see that while Dream Fields had worse results when using vanilla CLIP models for guidance, the LiT (Zhai et al., 2022) model used in their paper had a huge impact on making their results coherent. This suggests the fine-tuning process in LiT can be helpful in learning better features for guidance. We also find using alternative CLIP models such as OpenCLIP Ilharco et al. (2021) can result in high quality generation (see Appendix A.4). It may also be helpful to look into more recent text-image contrastive models such as SLIP (Mu et al., 2021) and UniCL (Yang et al., 2022). Our model can serve as a proxy qualitative measure for text-image similarity models to see how robust the features learned are and whether the models have learned underlying structures for objects that can be used to construct geometry that conforms with the real world.

**Dataset Supervision.** Another avenue to help with adversarial generations is to have a 3D dataset to help provide a good geometric prior for generators. Many current text to 3D works leverages this to generate higher quality results, albeit at the cost of text prompt freedom. Our work is orthogonal to theirs, and the techniques proposed in this paper can also be applied to their works.

**Resource Usage.** Overall, our models take less memory and are faster to train than Dream Fields (see Appendix A.5 for details). Our smallest model ($\text{Vox}_{\text{Exp}}$ with CLIP ViT-B/32) uses 7GB and can be optimized on a RTX 2080 Ti in roughly 20 minutes.

## 7 CONCLUSION

In this paper we have showed that it is possible to train an explicit voxel grid NeRF model with pure CLIP guidance to generate coherent results with a combination of image-based augmentations. We also provide empirical evaluations for how different augmentations as well as implicit voxel grids prevent adversarial examples within CLIP. Our model performs better than prior work when using the same CLIP model for guidance. It is also less compute and memory intensive while also being faster to train.

## 8  ETHICS STATEMENT

We acknowledge there are potential risks with using the CLIP (Radford et al., 2021) and Open-CLIP (Ilharco et al., 2021) models for guidance pertaining to deceptive and discriminatory content. CLIP was trained by OpenAI with an in-house dataset of 400 million text-image pairs. While the dataset is not publicly released, models that leverage CLIP for learning have demonstrated biases for applications like text-to-image generation (Struppek et al., 2022). OpenCLIP was trained on the LAION (Schuhmann et al., 2021) dataset which contains offensive and explicit images as shown by Birhane et al. (2021). Our model inherits these biases as we use these models as priors to generate 3D objects. On the one hand, text-to-3D models provide more accessible tools for the general public. On the other hand, they can also be used to produce harmful content by malicious actors to spread deceptive and hateful information. In addition, other potential ethical concerns raised by text-to-image (Nichol et al., 2021; Ramesh et al., 2022; Rombach et al., 2022; Saharia et al., 2022) models also apply. More systematic studies are needed in the space of text-to-3D models as higher quality generations are becoming possible and the potential risks need to be assessed and safeguards put in place to prevent misuse.

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

## A  APPENDIX

### A.1  HYPERPARAMETERS AND SCHEDULING

Our density and color MLPs are both 3-layer MLPs. The input dimension for both are $63$ which includes the raw coordinates and fourier features. The hidden feature size is $128$ for all layers and the output size is 1 and 3 for density and color MLPs respectively. The loss weight $\lambda$s for each of the terms can be found in Tab. 2. $\lambda_{\mathrm{CLIP}_2}$ is only set for models trained with two CLIP guidance models, otherwise it is set to $0$.

Table 2: Hyperparameters for the loss weights.

| Loss Weights | Value ($\mathrm{Vox_{Exp}}$) | Value ($\mathrm{Vox_{Imp}}$) |
|---|---|---|
| $\lambda_\sigma$ | 0.01 | 0.01 |
| $\lambda_{\mathrm{TV}}$ | 0.1 | 0.2 |
| $\lambda_{\mathrm{Tr}}$ | 0.5 | 0.5 |
| $\lambda_{\mathrm{KL}}$ | 0.05 | 0.2 |
| $\lambda_{\mathrm{CLIP}_2}$ | 0.5 | - |

The other important hyperparameters can be found in Tab. 3. Here the number of voxels decides the resolution of both the density and color grids. The grids are rectangular shapes with the proportions decided by the camera frustums and each side has integer number of voxels scaled so that the volume roughly has the amount of voxels as shown in the table. We also employ progressive scaling as in the original DVGO. For each scaling iteration the number of voxels is doubled with the starting number of voxels $\lfloor \frac{N_x \times N_y \times N_z}{2^{N_{\mathrm{pg}}}} \rfloor$, where $N_{\mathrm{pg}}$ is the number of progressive scaling iterations.

We find that it is important to schedule the $\mathcal{L}_{\mathrm{TV}}$ and $\mathcal{L}_{\mathrm{KL}_s}$ so that they are only turned on for some iterations. The TV loss $\mathcal{L}_{\mathrm{TV}}$ is applied only during the iterations defined in the table and turned off otherwise. Since the density values are near zero in the beginning, we find that the TV loss that encourages smoothness will prevent any densities from appearing. Also TV loss will make the object too smooth so we also turn it off towards the end to help the formation of sharp textures. For the KL loss $\mathcal{L}_{KL}$, we turn it on only in the beginning iterations. This is because the KL loss term helps to make the densities emerge, but may also encourage noisy densities appear outside the object within the sphere prior. For the 'CLIP ensemble iterations', it is only applied to the second CLIP guidance model $\mathcal{L}_{\mathrm{CLIP}_2}$ for ensembled guidance models.

Table 3: Hyperparameters for other values.

| Other Hyperparameters | Image Resolution | |
|---|---|---|
| | $168^2$ | $224^2$ |
| Training Iterations | 10k | 15k |
| Number of Voxels ($N_x \times N_y \times N_z$) | $140^3$ ($\mathrm{Vox_{Exp}}$) / $160^3$ ($\mathrm{Vox_{Imp}}$) | $160^3$ |
| Progressive Scaling Scheduling | $[4000, 6000, 8000]$ | $[5000, 7000, 9000, 11000]$ |
| TV Loss Iterations | $4000 \sim 9000$ | $5000 \sim 13000$ |
| KL Loss Iterations | $0 \sim 7000$ | $0 \sim 8000$ |
| CLIP Ensemble Iterations | $4000 \sim 10000$ | $5000 \sim 15000$ |

In Fig. 7 we illustrate the effects of both KL and TV losses. For KL loss we see a very clear circular hue around the generated object where the sphere prior locates. We turn this off in the later iterations to eliminate some noise densities around the object. For TV loss when it is turned on it helps to smooth the object. Again we turn it off towards the end of training. It can be seen that when it is turned off textures and edges appear more clearly to make the object look sharper.

### A.2  AUGMENTATION, ARCHITECTURE R-PRECISION

**Augmentation Ablation**  The R-Precision for augmentation ablations can be found in Tab. 4. Here $\mathrm{Vox_{Exp}}$ is trained using the ViT-B/16 CLIP model. Note that we add a variant of ViT-B/32 that was

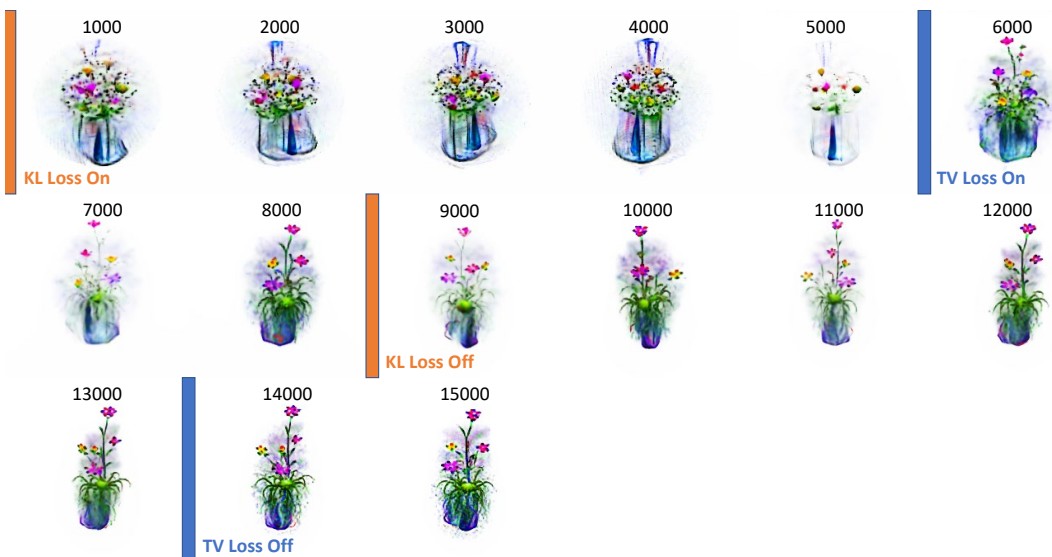

Figure 7: We show the rendered images from a random pose during the training process of our model with the prompt *Bouquet of flowers sitting in a clear glass vase.* and Vox$_{\text{Exp}}$ using ensemble ViT-B/32 + ViT-L/14 CLIP model as guidance. The number above image represents the training iterations. The orange bar and blue bar shows the duration $\mathcal{L}_{\text{KL}_s}$ and $\mathcal{L}_{\text{TV}}$ was turned on.

Table 4: We compare the R-Precision score for various augmentation configurations using different guidance models. The checks and crosses indicate which augmentations are turned on or off. R-Precision values in parentheses (middle column) indicate when the CLIP model used to evaluate retrieval is the same as the CLIP model used for optimization. We see that the DiffAug augmentation is important for high R-Precision and that PerspAug can hurt performance. Note that R-Precision is a fairly crude metric that only measures whether the text prompt can be retrieved from the generation and does not directly measure the quality of the generation.

| | Retrieval model R-Precision | | |
|---|---|---|---|
| Method (Diff/Back/Perspective) | ViT-B/32 | ViT-B/16 | ViT-B/32(OpenCLIP) |
| Vox$_{\text{Exp}}$ ($\times$/$\times$/$\times$) | 49.02 | (95.42) | 39.22 |
| Vox$_{\text{Exp}}$ ($\checkmark$/$\times$/$\times$) | 83.01 | (98.69) | 80.39 |
| Vox$_{\text{Exp}}$ ($\times$/$\checkmark$/$\times$) | 49.67 | (91.50) | 45.10 |
| Vox$_{\text{Exp}}$ ($\checkmark$/$\checkmark$/$\times$) | 84.97 | (98.69) | 92.16 |
| Vox$_{\text{Exp}}$ ($\checkmark$/$\checkmark$/$\checkmark$) | 83.01 | (90.85) | 84.97 |

trained on the LAION-2B (Schuhmann et al., 2021) dataset to the evaluation, we indicate this model as ViT-B/32(OpenCLIP). We see from the results when we add DiffAug, the R-Precision value goes up significantly. This indicates the texture of the object plays a large role in the coherency of the object according to CLIP. When just adding BackAug, the scores do not go up by much. However, when we add both DiffAug and BackAug the R-Precision scores are the best. PerspAug decreases the R-Precision score when added, this is probably due to the reduction in texture sharpness we saw in Fig. 4. However, qualitatively for some text prompts it helps to remove noise and unwanted background textures making the object have better geometric coherency. We notice that for many objects with broken geometry as long as the texture resembles the text, the R-Precision will still be high. This is a big limitation of CLIP R-Precision as it is not a good metric for whether or not the generated object has good geometric structure.

**Architecture Ablation** The R-Precision for Vox$_{\text{Exp}}$ and Vox$_{\text{Imp}}$ trained with different CLIP architectures can be found in Tab. 5. For Vox$_{\text{Exp}}$ all augmentations are turned on, for Vox$_{\text{Imp}}$ DiffAug and BackAug are turned on. We refer to the ViT-B/32(OpenCLIP) column to compare methods as the models are trained either using ViT-B/32 or ViT-B/16. The Vox$_{\text{Exp}}$ model trained with ViT-B/16

Table 5: Here we calculate the R-Precision for different guidance models that were used to train $\text{Vox}_{\text{Exp}}$ and $\text{Vox}_{\text{Imp}}$. R-Precision values in parentheses indicate when the CLIP model used to evaluate retrieval is the same as the CLIP model used for optimization.

| | Retrieval model R-Precision | | |
|---|---|---|---|
| Method (CLIP) | ViT-B/32 | ViT-B/16 | ViT-B/32(OpenCLIP) |
| $\text{Vox}_{\text{Exp}}$ (ViT-B/32) | (87.58) | 73.20 | 80.39 |
| $\text{Vox}_{\text{Exp}}$ (ViT-B/16) | 83.01 | (90.85) | 84.97 |
| $\text{Vox}_{\text{Exp}}$ (ViT-B/32 + ViT-L/14) | (71.90) | 61.44 | 61.44 |
| $\text{Vox}_{\text{Imp}}$ (ViT-B/32) | (99.34) | 90.20 | 94.12 |
| $\text{Vox}_{\text{Imp}}$ (ViT-B/16) | 88.24 | (98.69) | 85.62 |

*"Apples on tree ready to pick in garden area."*

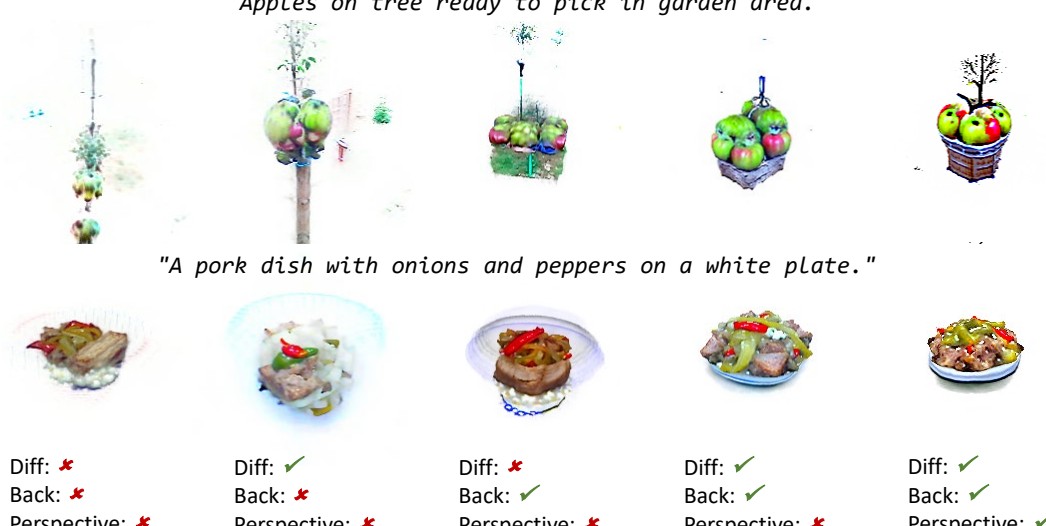

*"A pork dish with onions and peppers on a white plate."*

| Diff: ✗ | Diff: ✓ | Diff: ✗ | Diff: ✓ | Diff: ✓ |
| Back: ✗ | Back: ✗ | Back: ✓ | Back: ✓ | Back: ✓ |
| Perspective: ✗ | Perspective: ✗ | Perspective: ✗ | Perspective: ✗ | Perspective: ✓ |

Figure 8: Augmentation ablation for the $\text{Vox}_{\text{Imp}}$ model. The implicit model is able to produce results without unwanted background textures even without perspective augmentation.

has the highest score for explicit models. We see that the ensemble model (ViT-B/32+ViT-L/14) has much lower scores compared to the other explicit models. Qualitatively, the ensemble models have higher texture details for most objects. However, we also notice that for some prompts, subjects mentioned in the input text are not present in the generated object of the ensemble model that was otherwise present when just using ViT-B/32 or ViT-B/16 for guidance. For $\text{Vox}_{\text{Imp}}$, the model trained with ViT-B/32 had the highest performance. Qualitatively, objects generated by $\text{Vox}_{\text{Imp}}$ trained with ViT-B/16 can generate more detailed geometry. In Dream Fields, their ViT-B/32 model also outperformed their ViT-B/16 model in terms of R-Precision.

**CLIP R-Precision** While the R-Precision is correlated with the faithfulness of the generated object with respect to the prompt, higher scores do not always yield the best qualitative results. The R-Precision score itself is also prone to adversarial examples. This suggests that quantitative evaluation for Text-to-3D models is still very challenging and better metrics are required.

## A.3 VOX$_{\text{IMP}}$ AUGMENTATION ABLATION

We show the augmentation ablation for the $\text{Vox}_{\text{Imp}}$ model with KL loss on in Fig. 8. Overall we find that compared to $\text{Vox}_{\text{Exp}}$, the $\text{Vox}_{\text{Imp}}$ model is less noisy even in cases where less augmentations are on. There seem to be no unwanted background textures even when just Diff and Back augmentations are on. Since the perspective augmentation can sometimes hurt the texture sharpness we turn it off for $\text{Vox}_{\text{Imp}}$ as it seems unnecessary.

## A.4 OpenCLIP Model Results

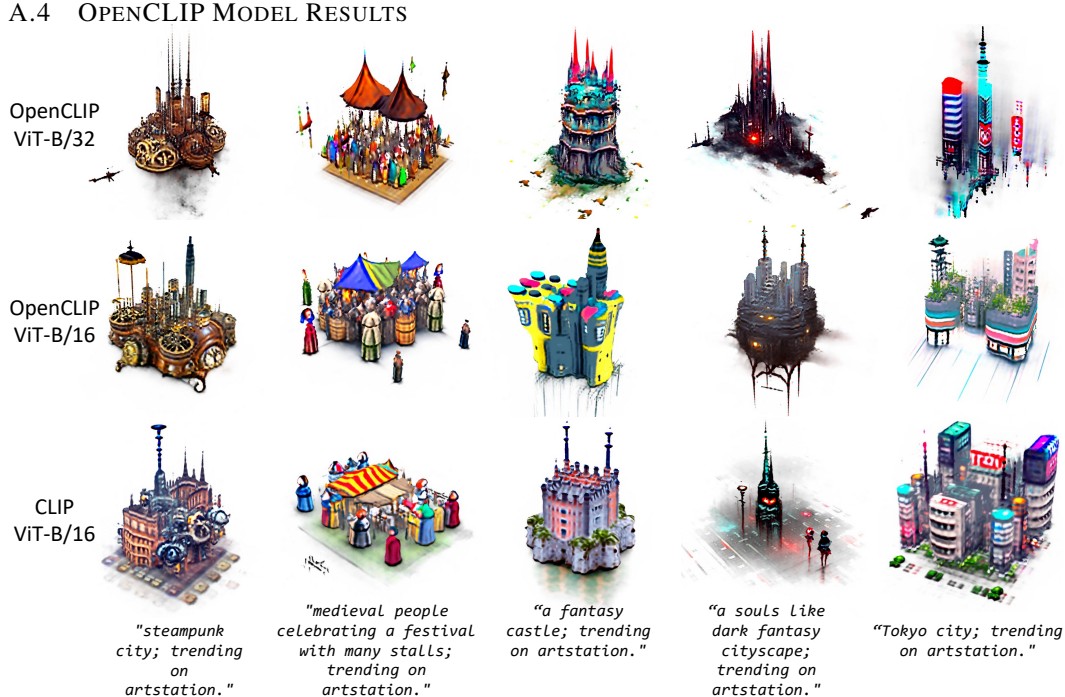

OpenCLIP
ViT-B/32

OpenCLIP
ViT-B/16

CLIP
ViT-B/16

*"steampunk city; trending on artstation."*  *"medieval people celebrating a festival with many stalls; trending on artstation."*  *"a fantasy castle; trending on artstation."*  *"a souls like dark fantasy cityscape; trending on artstation."*  *"Tokyo city; trending on artstation."*

Figure 9: In this figure we test out OpenCLIP's model trained on the LAION dataset as guidance with the Vox$_{\text{Imp}}$ model. We can see that there is not one model that performs the best across all the prompts.

In this section we tested out another implementation of CLIP, namely OpenCLIP Ilharco et al. (2021) which was trained on the LAION Schuhmann et al. (2021) dataset. Particularly we use the ViT-B/32 variant trained on LAION-2B and the ViT-B/16 variant trained on LAION-400M. We use the Vox$_{\text{Imp}}$ model and the image resolution is set to $224^2$ during training. Our results are shown in Fig. 9. Overall, we see that there is not one model that performed the best across the board. For some prompts, all three models generated coherent results and the variation between them can be desirable for creative applications. It would be interesting to explore further how varying datasets as well as size can have an impact on the quality and diversity of the generated results.

## A.5 Resource Usage

In Tab. 6 the memory usage and time consumption is shown across the devices we conducted training on. All of our model variations consumed less memory compared to Dream Fields and was faster to train. Their model takes a lot of memory because of the 8-layer MLP with residual connection used for the NeRF model. Since we mostly used RTX A6000s with 48 GB of VRAM we didn't optimize hyperparameters such as voxel grid resolution much unless we ran out of memory. For code release we will provide both Vox$_{\text{Exp}}$ and Vox$_{\text{Imp}}$ model settings that will run on more commodity GPUs. It is worth noting that the Vox$_{\text{Imp}}$ model trained at image resolutions $168^2$ and $224^2$ consumes about the same amount of memory. This is because we evaluate the forward pass on the entire $\boldsymbol{V}^{(\text{PE})}$ grid and since both variations has the same voxel grid resolution their memory consumption is about the same. We plan to update this in the future so that regions of the voxels that are not hit by rays will be masked out from the calculation of the forward pass. We were also not able to run Dream Fields at higher resolutions in their paper because of availability issues with TPU v3-8.

Table 6: In this table we show the memory consumption as well as training time per prompt for various resolution and training iteration settings in comparison with Dream Fields across the devices that we used.

| Method | Res + Iter | Device | Memory | Time |
|---|---|---|---|---|
| Vox$_{\text{Exp}}$ (ViT-B/32) | $168^2$ + 10k | RTX 2080 Ti | 7.01 GB | $\sim$ 21 min |
| Vox$_{\text{Exp}}$ (ViT-B/32) | $224^2$ + 15k | RTX 2080 Ti | 9.76 GB | $\sim$ 35 min |
| Vox$_{\text{Exp}}$ (ViT-B/16) | $168^2$ + 10k | RTX A6000 | 12.39 GB | $\sim$ 42 min |
| Vox$_{\text{Exp}}$ (ViT-B/16) | $224^2$ + 15k | RTX A6000 | 14.92 GB | $\sim$1 hr 6 min |
| Vox$_{\text{Exp}}$ (ViT-L/14) | $224^2$ + 15k | RTX A6000 | 21.48 GB | $\sim$ 1 hr 54 min |
| Vox$_{\text{Exp}}$ (ViT-B/32+ViT-L/14) | $224^2$ + 15k | RTX A6000 | 33.06 GB | $\sim$1 hr 40 min |
| Vox$_{\text{Exp}}$ (ViT-B/16+ViT-L/14) | $224^2$ + 15k | RTX A6000 | 35.37 GB | $\sim$ 1 hr 54 min |
| Vox$_{\text{Imp}}$ (ViT-B/16) | $168^2$ + 10k | RTX A6000 | 41.39 GB | $\sim$ 40 min |
| Vox$_{\text{Imp}}$ (ViT-B/16) | $224^2$ + 15k | RTX A6000 | 40.75 GB | $\sim$ 1 hr 4 min |
| Dream Fields (ViT-B/16) | $168^2$ + 10k | RTX A6000 | OOM ($>$ 48 GB) | - |
| Dream Fields (ViT-B/16) | $168^2$ + 10k | TPU v2-8 | - | $\sim$2 hr 17 min |

