# OpenReview forum: "UNDERSTANDING PURE CLIP GUIDANCE FOR VOXEL GRID NERF MODELS"
_ICLR.cc/2023/Conference — Submitted to ICLR 2023_

### Official Review · Reviewer_XGe4 · 2022-10-17

**Confidence:** 4
**Correctness:** 4
**Technical Novelty And Significance:** 2
**Empirical Novelty And Significance:** 3
**Recommendation:** 6

**Clarity, Quality, Novelty And Reproducibility:**

The paper is clearly written. Implementation details, running speed, and memory cost are provided in Appendix. Thus I think it is reproducible. The 3D synthesis results obtained in this work is very promising. The qualitative examples shown are significantly better than prior works. However, the novelty is limited as this work mainly focus on a thorough study of several design choices.

**Details Of Ethics Concerns:**

The synthesized results from the proposed method may contain bias due to the bias in the dataset used for training CLIP.

**Strength And Weaknesses:**

Strength:

- This paper presents a systematic study on several design choices for pure clip guided text to 3D NeRF generation, which has not been done before.  In general, the study summarizes several techniques that would stabilize training and prevent adversarial results. The experimental results would serve as a useful guidance in related tasks.

- Combining several designs, the final model achieves impressive 3D grid synthesis using CLIP guidance only. Compared to DreamField, it achieves more coherent results with higher memory efficiency and faster training speed.


Weaknesses:

- As a work aiming for "systematic study", the quantitative study is weak. Quantitative results are only provided for the final models but not for ablation study. It is unclear how consistent is the ablation study as only a few qualitative examples are provided. Thus, I suggest also providing quantitative results for the ablation studies on data augmentations and backbones.

- The design choices studied in the paper (different augmentations, backbones, voxel grids) are simply some common techniques that exist before and there is not much new techniques in the paper. The results are also not surprising, as it is intuitive that augmentations, model ensemble, and implicit fields can improve results.

- (Minor) For the implicit model, it is not explained why it is used to model only grid coordinates instead of continuous coordinates like NeRF. My guess is that this is more efficient, but it's better to mention it.

- Typo:
Page 5 line 7: ',' -> '.'
Section 6 line 7: 'effect' -> 'affect'

**Summary Of The Paper:**

This paper presents a systematic study on several design choices for pure clip guided text to 3D NeRF generation. It shows how different image augmentations improves the results. Different CLIP architectures are compared and it is observed that the ensemble of CLIP models performs better. Authors also show that implicit grid model parameterized by MLPs perform better than explicit grid. Additionally, this work introduces a spherical prior that facilitate training in this task. The final model achieves satisfactory grid synthesis using CLIP guidance only, which outperforms previous methods.

**Summary Of The Review:**

This paper, by studying a number of design factors, finds a good setting that achieves impressive 3D object synthesis from CLIP guidance only. The performance is much better than previous methods. However, the technical novelty is somewhat limited and the quantitative study is weak. Considering the impressive results and that this work focus on a systematic study, the novelty is not a big issue and I am inclined to accept this paper. But I suggest adding more quantitative evaluation to make the results more solid.

---

> ### Author Response · Authors · 2022-11-19
> **Response to Reviewer XGe4**
>
> > “As a work aiming for "systematic study", the quantitative study is weak. Quantitative results are only provided for the final models but not for ablation study. It is unclear how consistent is the ablation study as only a few qualitative examples are provided. Thus, I suggest also providing quantitative results for the ablation studies on data augmentations and backbones.”
>
> * We have added a quantitative study of the data augmentation and backbones to the paper. Please refer to the Appendix section A.2 for these results.
>
> > “(Minor) For the implicit model, it is not explained why it is used to model only grid coordinates instead of continuous coordinates like NeRF. My guess is that this is more efficient, but it's better to mention it.”
>
> * Thanks for the constructive feedback. It is as you have guessed, we predict density and color values on the grid coordinates for memory efficiency. As we also keep track of a mask for voxels with low densities, we can easily skip predicting the color and density values of these vertices which reduces the number of forward passes through the MLP.
>
> > “Typo:”
>
> * Fixed.
>
> > “Ethics Concerns”
>
> * We have added an Ethics Statement to our paper. Please refer to section 8 of the revision.

---

### Official Review · Reviewer_3Rq7 · 2022-10-19

**Confidence:** 4
**Correctness:** 4
**Technical Novelty And Significance:** 2
**Empirical Novelty And Significance:** 2
**Recommendation:** 5

**Clarity, Quality, Novelty And Reproducibility:**

This writing and organization of this paper is great. As I said in the last section, the key technical contributions of this paper is not very clear to me and makes me feel like reading a technique report.

The paper is reproducible since it provides comprehensive details for implementation.

**Strength And Weaknesses:**

Strength:
- This paper provides a systematic study of the design choices towards the text-to-3D methods, which I think may guide the future work in this topic
- The authors propose a novel spherical prior for optimization
- By applying the strategies the authors studies, this paper achieves state-of-the-art results on text-to-3D synthesis

Weaknesses:
As a technical paper, I think this paper might lack enough technical contributions, which makes my consider this a "benchmark paper". However, in this case, I found the studied a fairly narrow topic - only text-to-3D with CLIP and only on a few aspects like augmentation, CLIP architecture. I think there are many other aspects the authors might consider: synthesizing meshes as included in the related work, other feature encodings like instant-ngp. Also, I am not sure why the authors didn't use the Plenoxel model, which is also a discrete voxel representation.

**Summary Of The Paper:**

This paper tackles the problem of using CLIP for text-guided 3D shape generation. The authors experimented on different image-based augmentations, different CLIP backbones and analyzed their influences toward final results. They also studies the regularization effects of explicit vs. implicit voxel grid models. Results shown that by combining those design considerations, this paper demonstrated better visual quality with higher memory efficiency and less training time.

**Summary Of The Review:**

I think this paper offers a great study for text-to-3D synthesis by studying the CLIP architecture, 3D representation and image augmentation, and the results outperforms quite a lot upon previous methods. The only thing I am skeptical is the technique contributions and overall contribution to the research community. Therefore, currently I think currently it is below the acceptance threshold. But if the authors provides more clarification on my questions, I am willing to discuss and change my rating.

---

> ### Author Response · Authors · 2022-11-19
> **Response to Reviewer 3Rq7**
>
> > “As a technical paper, I think this paper might lack enough technical contributions, which makes my consider this a "benchmark paper". However, in this case, I found the studied a fairly narrow topic - only text-to-3D with CLIP and only on a few aspects like augmentation, CLIP architecture. I think there are many other aspects the authors might consider: synthesizing meshes as included in the related work, other feature encodings like instant-ngp. Also, I am not sure why the authors didn't use the Plenoxel model, which is also a discrete voxel representation.”
>
> * We agree that it is worth exploring these aspects for synthesizing meshes and whether meshes also provide an extra layer of regularization compared to NeRF models.  In this paper, our focus is mainly on augmentation strategies and implicit vs explicit voxel NeRF representation choices.
> * Learnable feature encodings like the ones used by instant-ngp allow for better modeling of complicated objects in the reconstruction task. However, in our setting where CLIP is used as the objective the main concern is preventing adversarial generations. Having a NeRF model with more capacity such as learnable feature encodings can actually make this problem worse as there are more parameters to optimize and the model can potentially more easily find adversarial examples for CLIP. Most CLIP models are trained using image resolution of 224x224, so having more powerful representation might not help as we are also limited by details displayable in this resolution.  In our results, we have found that our current NeRF capacity is not a limiting factor and can learn on a wide variety of different text prompts.
> * For Plenoxel, there are mainly two differences compared to our explicit model: 1) Sparse Voxel Grids and 2) Spherical Harmonics to represent color. For the first point, Plenoxel only stores features for occupied voxels in an array and ignores unoccupied voxels during rendering. Our backbone DVGO model also ignores voxels that have low density values. This aspect is functionally equivalent to Plenoxels so we do not expect it to impact results. The main advantage of point 2 (Spherical Harmonics) is ability to model view-dependent RGB values. We have tried using Spherical Harmonics for the color voxel grid and found they do not produce obvious differences compared to predicting RGB values directly. We believe this is due to CLIP not being sophisticated enough to model view-dependent appearance as it is trained on single-view images.

---

### Official Review · Reviewer_hRKS · 2022-10-24

**Confidence:** 4
**Correctness:** 3
**Technical Novelty And Significance:** 2
**Empirical Novelty And Significance:** 2
**Recommendation:** 5

**Clarity, Quality, Novelty And Reproducibility:**

Clarity, Quality and Reproducibility is ok.
The Novelty is limited.

**Strength And Weaknesses:**

Strength:
This paper is well written.

Weaknesses:
1. The technical contribution is very limited in this work.
They explores the effect of different augmentations, clip backbone, scene representation forms, which may contribute to community.
However, only simply giving the conclusion & findings is not enough, some deeper analysis and theoretical explanation for the findings are also expected, or the paper will tend to be an experiment report instead of research paper. For instance, why KL loss is essential when augmentations are turned on in Fig. 3?

2. Some implementation is not clear. For the KL divergence computation, eq. (4) and (5) are given but I cannot understand eq. (5) for the KL divergence computation (if the formulation is correct?) and it is also not clear how (5) is computed or optimized.

3. The improvement compared with Dream Fields tends to be incremental as shown in Fig. 6.

**Summary Of The Paper:**

This work conducts a systematic study of augmentations and their effect on text to 3D generation results with pure CLIP guidance.
They compare different CLIP backbones for guidance as well as model ensembles for finer 3D object detail, and compare the regularization effects on geometry of explicit vs implicit voxel grids.
They demonstrate generation of high-resolution grids using CLIP guidance only.

**Summary Of The Review:**

As mentioned above, some deeper analysis and theoretical explanation are expected for the experiment results.
Some implementation is not clear.

---

> ### Author Response · Authors · 2022-11-19
> **Response to Reviewer hRKS**
>
> > “The technical contribution is very limited in this work. They explores the effect of different augmentations, clip backbone, scene representation forms, which may contribute to community. However, only simply giving the conclusion & findings is not enough, some deeper analysis and theoretical explanation for the findings are also expected, or the paper will tend to be an experiment report instead of research paper. For instance, why KL loss is essential when augmentations are turned on in Fig. 3?”
>
> We appreciate the constructive comments. Here we provide some additional explanations and insight for why the augmentations and KL loss are needed:
>
> * CLIP similarity only maximizes matching of text to the generated content, there is no guarantee that the generation has coherent 3D structure. The various augmentation schemes and KL loss act as regularizers to ensure more coherent 3D structure emerges.
> In order to optimize the model parameters, smooth gradients from the similarity loss function are needed. However, we speculate that when augmentations are turned on, the gradients become non-continuous in the beginning as the model might have to modify multiple voxels in the scene in a particular way for the similarity loss to go down. This makes the model difficult to train.  We see evidence of this in near zero gradient magnitudes when augmentations are turned on without KL loss (see Fig. 3, 2nd and 3rd column).  In contrast, when the augmentations are off, the model can more easily find examples allowing for smoother gradients (see non-zero gradients in Fig. 3, 1st column).  However, while the model can train, the examples it finds that increase the CLIP similarity are more likely to be adversarial examples that do not have coherent structure.  By adding the spherical KL loss term, we help regularize the 3D structure that is generated to be more compact and coherent.  This provides a gradient during the beginning of training and helps the densities to emerge and a meaningful shape to coalesce (see Fig. 3, 4th to 6th column).
> * For Diff Augment, we can see that the generated objects have sharper textured appearance. We attribute this to the color jittering in the augmentation which modifies the brightness, saturation and contrast of the image. This could help the model generate higher fidelity colors for the final result.
> * Without background augmentations the densities in the scene are sparsely distributed with empty spaces in between. Adding the background augmentation causes the densities to be mixed up with the background if they are too far apart. This could help the object form more compact densities that actually seem like a continuous object instead of point cloud-like densities.
> * Finally using the perspective augmentation helps imitate rendering of the object at different camera poses from one image. This could help eliminate noise or unwanted background textures as viewing it from a different perspective might result in lower CLIP similarity scores.
>
> > “Some implementation is not clear. For the KL divergence computation, eq. (4) and (5) are given but I cannot understand eq. (5) for the KL divergence computation (if the formulation is correct?) and it is also not clear how (5) is computed or optimized.”
>
> * We have  revised eq. (5) to clarify. Here $\alpha^{(post)}(q,V^{(density)})$ is the density value from our density voxel grid and $P_{sphere}(q)$ is the spherical prior function defined in eq. (4). We calculate the KL divergence between these two values for all coordinates $q$ that belong to the grid vertices of our voxel grid.
>
> > “The improvement compared with Dream Fields tends to be incremental as shown in Fig. 6.”
>
> * The qualitative improvements may seem incremental as we are only able to show static images of the generated results in the paper. However, if viewed from multiple views our results have higher quality compared to Dream Fields. Please refer to our supplementary website: https://isekaicoder.github.io/ICLR3801-Supplemental/ for multi-view results of the objects compared to Dream Fields. In terms of memory and training time per prompt we are also more efficient compared to Dream Fields as can be seen in Tab. (7).

---

> > ### Comment · Reviewer_hRKS · 2022-12-12
> > **Thanks for your response**
> >
> > Thank you for taking the time to clarify my questions. I have read the author's response and also other reviews. My main concern is still the technical contributions of this work, which means major revision should be performed for this work. Thus, I keep my original rating.

---

### Author Response · Authors · 2022-11-19
**General Comment by Paper3801 Authors**

We thank all of the reviewers for taking the time to read our paper and provide valuable feedback. We respond to individual reviewer questions in sub-threads, and welcome any additional questions.

Here, we briefly reiterate our contributions. We investigate the nascent field of using CLIP models to guide 3D generation from text. While different augmentations have been proposed by prior work, there has been no systematic study of how individual augmentations impact training and performance of text to 3D generation models, and how they might be effectively combined. In addition, we demonstrate that it is possible to have a light and efficient explicit voxel grid representation that can be trained on a single commodity GPU. This enables members of the research community with limited computational resources to study open vocabulary generation of 3D content.

To address reviewer feedback, we have also updated the paper as follows:
- Revised Eq (5) so it is more clear
- Added Appendix A.2 with quantitative study of the effect of data augmentations and backbones
- Statement about potential ethical issues with CLIP guidance for text to 3D generation

---

> ### Author Response · Authors · 2022-12-03
> **We welcome additional feedback**
>
> We thank all the reviewers once again for their time and constructive comments.  We hope we have addressed reviewer concerns.  Please let us know if you have any remaining questions or concerns that we can clarify during the discussion period.

---

### Decision · Program_Chairs · 2023-01-20

**Decision:**

Reject

**Justification For Why Not Higher Score:**

Not enough technical contribution/insight.

**Justification For Why Not Lower Score:**

N/A

**Metareview: Summary, Strengths And Weaknesses:**

The reviewers agree that the paper provides a well executed, and well described, set of experiments that shed light on behavior of "CLIP guidance" for 3D model generation. However, the reviewers also agree (this includes the sole barely positive review) that the technical contributions are very limited. In particular, the analysis of the possible reasons for the findings in the submission is unsatisfying. The scope of the experiments is also somewhat limited (this on its own would be less of a problem).

Overall, this submission is to me clearly below acceptance threshold. I hope the authors continue their study, and resubmit in the future with more analysis. (With the caveat that the field of text-to-3D seems to have coalesced around an alternative approach, not involving CLIP guidance, namely conditional diffusion models, but of course it's far from settled and I think there is plenty of room for better understanding of CLIP based methods.)